# Cohort profile: the multigeneration Respiratory Health in Northern Europe, Spain and Australia (RHINESSA) cohort

Cecilie Svanes,[1,2] Ane Johannessen,[2] Randi Jacobsen Bertelsen,[3,4] Shyamali Dharmage,[5] Bryndis Benediktsdottir,[6,7] Lennart Bråbäck,[8] Thorarinn Gislason,[7] Mathias Holm,[9] Oskar Jõgi,[3,10] Caroline J Lodge,[11] Andrei Malinovschi,[12] Jesus Martinez-Moratalla,[13,14] Anna Oudin,[8] José Luis Sánchez-Ramos,[15] Signe Timm,[16,17] Christer Janson ![ORCID],[12,18] Francisco Gomez Real,[3,19] Vivi Schlünssen,[20,21] on behalf of the RHINESSA International Collaboration

CS and AJ are joint first authors.

**Correspondence to**
Professor Cecilie Svanes;
Cecilie.Svanes@uib.no

## ABSTRACT

**Purpose** The Respiratory Health in Northern Europe, Spain and Australia (RHINESSA) cohort was established to (1) investigate how exposures before conception and in previous generations influence health and disease, particularly allergies and respiratory health, (2) identify susceptible time windows and (3) explore underlying mechanisms. The ultimate aim is to facilitate efficient intervention strategies targeting multiple generations.

**Participants** RHINESSA includes study participants of multiple generations from ten study centres in Norway (1), Denmark (1), Sweden (3), Iceland (1), Estonia (1), Spain (2) and Australia (1). The RHINESSA core cohort, adult offspring generation 3 (G3), was first investigated in 2014–17 in a questionnaire study (N=8818, age 18–53 years) and a clinical study (subsample, n=1405). Their G2 parents participated in the population-based cohorts, European Community Respiratory Heath Survey and Respiratory Health In Northern Europe, followed since the early 1990s when they were 20–44 years old, at 8–10 years intervals. Study protocols are harmonised across generations.

**Findings to date** Collected data include spirometry, skin prick tests, exhaled nitric oxide, anthropometrics, bioimpedance, blood pressure; questionnaire/interview data on respiratory/general/reproductive health, indoor/outdoor environment, smoking, occupation, general characteristics and lifestyle; biobanked blood, urine, gingival fluid, skin swabs; measured specific and total IgE, DNA methylation, sex hormones and oral microbiome. Research results suggest that parental environment years before conception, in particular, father's exposures such as smoking and overweight, may be of key importance for asthma and lung function, and that there is an important susceptibility window in male prepuberty. Statistical analyses developed to approach causal inference suggest that these associations may be causal. DNA methylation studies suggest a mechanism for transfer of father's exposures to offspring health and disease through impact on offspring DNA methylation.

**Future plans** Follow-up is planned at 5–8 years intervals, first in 2021–2023. Linkage with health registries contributes to follow-up of the cohort.

## Strengths and limitations of this study

⇒ The main strength of the Respiratory Health in Northern Europe, Spain and Australia (RHINESSA) cohort is the availability of rich preconception exposure information for a large number of young adolescent and adult study participants, from both the paternal and maternal line, taking advantage of extensive information collected from mothers/fathers over 20 years of childbearing age.

⇒ Excellent health and population registries in the Northern European study centres contribute to unbiased identification of study participants and enrichment of data, and, for some study centres, provide additional information on multiple generations covering cohorts born over the last century.

⇒ The multigeneration design and harmonisation of study protocols across generations provide a valuable opportunity to validate next of kin information, thereby improving the validity of retrospectively collected data on family members.

⇒ The Spanish and Australian study centres contribute to generalisability beyond Northern Europe which has the majority of study participants, however, generalisation to low-income countries must be done with care.

⇒ Weaknesses of RHINESSA further include relatively low response rates, partly mitigated by the opportunity to analyse selection bias based on parental data for responders and non-responders; further, extensive exposure data is only available from one parent in most study centres, while information on the other parent is available from next of kin data reported by the offspring, and from registry data in the Nordic study centres.

## INTRODUCTION

While it is generally acknowledged that intrauterine life and early childhood is essential to health and disease throughout life, emerging evidence supports that there may

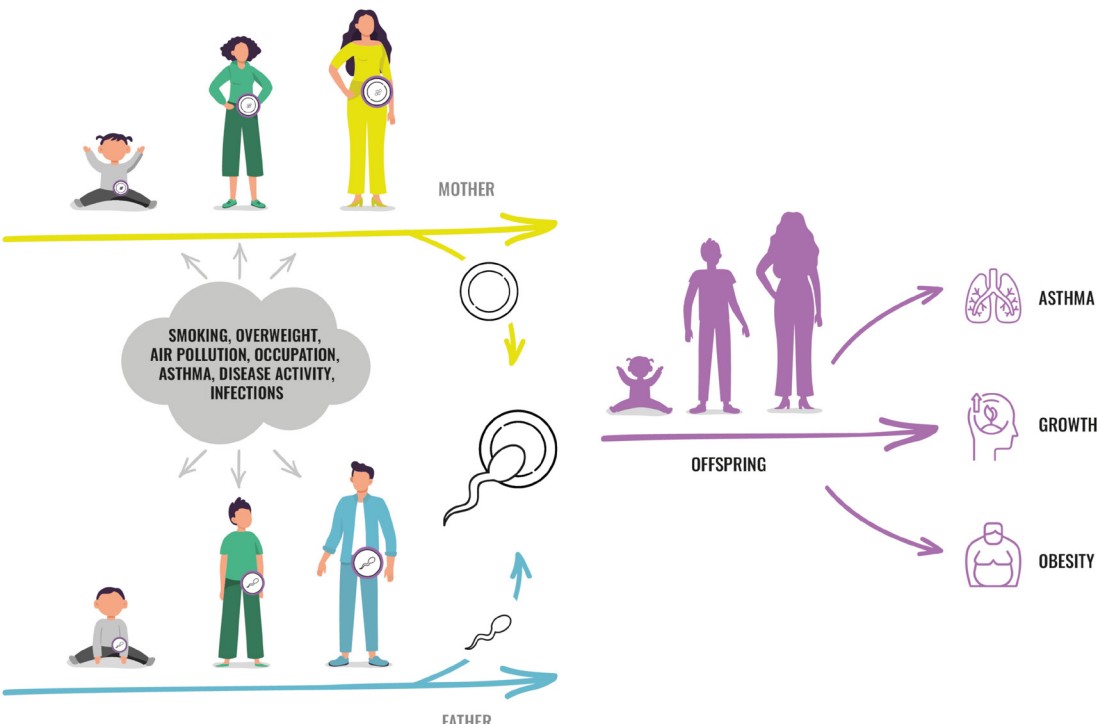

**Figure 1** RHINESSA study concept. The RHINESSA multigeneration study provides data and biomaterial to study how factors in girls and boys, during different age windows, can influence the health of their future children. factors such as smoking, overweight and air pollution could influence the developing and maturing germ cells in both sexes, and a 'fingerprint' of such exposures could be transferred to future offspring and thereby influence their phenotype. RHINESSA, Respiratory Health in Northern Europe, Spain and Australia.

be important susceptibility windows before conception.[1–7] The hypothesis arose from new understanding of epigenetic mechanisms by which environmental effects could be transferred across generations[2 8–10] and from studies supporting that such transfer of non-mutagenic environmental effects across generations could actually be taking place in humans.[11–13] Theoretically, an exposure affecting one person might at the same time affect that person's germ cells, and thereby the health of future offspring (figure 1). The intrauterine period and male puberty may be time windows when the germ cells are more susceptible to external and internal factors due to extensive epigenetic reprogramming.[2 6 14]

Knowledge on the early life origins of health and disease led to a paradigm shift in public health strategies, and is today implemented in public health programmes targeting mother and child across the globe. The concept of preconception origins of health and disease, of susceptible time windows before conception, opens a new perspective on public health: Are there opportunities for preventive measures that may result in improved health, not only for the individual but also for their future offspring and generations?[15]

There is a need to establish human generation cohorts that are tailored to investigate the preconception origins of health and disease. Most available literature is based on animal studies. There are human cohort studies with preconception data, such as for example, the Isle of Wight Studies, the Avon Longitudinal Study of Parents and Children study, the Lifelines NEXT generation study, and the Tasmanian Longitudinal Health Study. However, birth cohort studies often have not collected data from the fathers, or from the childhood/adolescence of any of the parents. Since the human reproductive cycle spans decades, investigating exposure effects from before conception and across generations represents a great challenge.

The RHINESSA study is designed to address this by investigating the offspring of persons who have already been extensively characterised during 20 years of their reproductive life. RHINESSA is an international multigeneration multicentre study established to research the preconception origins of health and disease, in particular allergies and respiratory health. The aims of RHINESSA are to investigate the influence of exposures before conception including in previous generations for health and disease, to identify potentially susceptible time windows for such influences, and to explore mechanisms for exposure effects. RHINESSA's primary focus is allergies and chronic respiratory disorders, namely asthma, rhinoconjunctivitis, allergic sensitisation, eczema, chronic obstructive pulmonary disease, lung function and sleep disorders. The cohort resource and research methodologies of RHINESSA also have the capacity for multigeneration research in other areas, such as obesity, women's health and oral health. The ultimate aim of RHINESSA is to improve health at large by generating a knowledge base for efficient strategies that may improve health over several generations.

# MULTIGENERATIONAL STUDY RHINESSA
## ADULT OFFSPRING COHORT

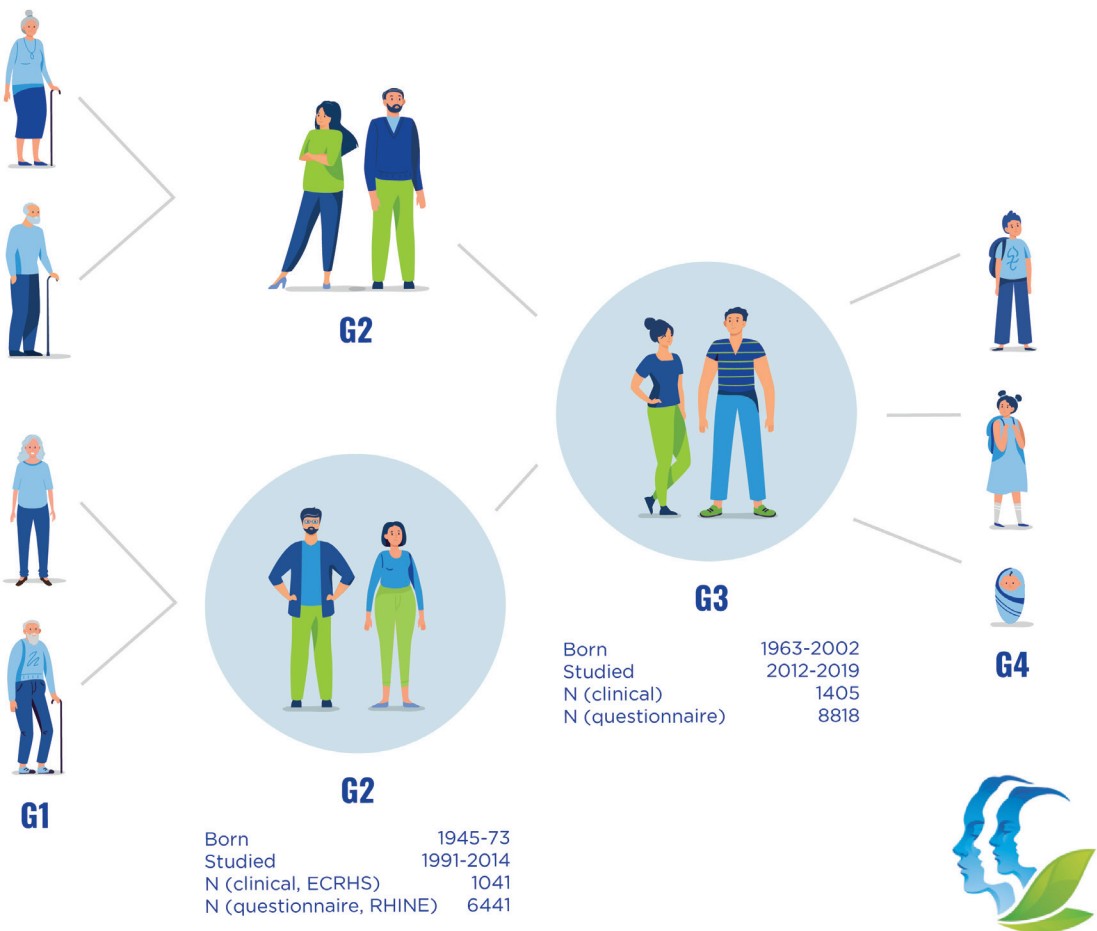

**Figure 2** RHINESSA study design. The RHINESSA adult offspring cohort (generation 3 'G3') includes 8818 young men and women investigated with questionnaires (q), of which 1405 were investigated clinically (c). These are the offspring of men and women participating in the RHINE/ECRHS studies (G2) who were followed up over 20 years. In addition, Aarhus, Bergen, Melbourne and Tartu study centres investigated G3) offspring age 4–17 years (1139q/ 201 c), and Bergen study centre investigated G1) (1470q/145 c), the other G2 parent (910q/152 c) and G4) (750q/433 c). In all study centres G3 and G2 study participants provided information about their parents and offspring in G1 and G4. ECRHS, European Community Respiratory Heath Survey; RHINESSA, Respiratory Health in Northern Europe, Spain and Australia.

## COHORT DESCRIPTION

This cohort profile describes the RHINESSA adult offspring cohort (generation 3 (G3) of ≥18 years of age) and their G2 parents investigated as part of the European Community Respiratory Health Survey (ECRHS) and Respiratory Health in Northern Europe (RHINE) studies (figure 2). Online supplemental file 1 gives summary data for younger offspring and additional cohorts (G1–G4) investigated in some study centres—altogether four generations.

## Study design

RHINESSA builds on the large longitudinal studies of respiratory health in adults, the ECRHS, www.ecrhs.org) established in the early 1990's[16–18] and the linked study, RHINE, www.rhine.nu).[19 20] For a range of environmental exposures and lifestyle factors, the ECRHS and RHINE cohorts (G2) have data with high time-resolution, both before and during the age of childbearing. The children born to these parents, are the target population of RHINESSA (G3). Bergen RHINESSA also investigated the G1 grandparent generation, the G2 parent not participating in RHINE/ECRHS, and the G4 offspring's offspring. Summary data for these additional cohorts are given in online supplemental table S2. Northern Europe is well suited for generation studies due to excellent national registries with full coverage of the populations for decades, providing means to identify family members in an unbiased manner as well as information on exposures and outcomes (ie, (life-time) home addresses for geocoding, prescription registries for asthma

Table 1 Sources of identification of RHINESSA adult participants (18+ years) (G3) and their parents (G2), by centre, including a questionnaire cohort (8818 offspring with their 6441 parents), and a clinical cohort (1405 offspring with their 1041 parents)

| Study centre | Parents (G2) | | RHINESSA adult offspring (G3) | |
| --- | --- | --- | --- | --- |
| | Source used for identifying offspring | N | Source of identification | N included in cohort |
| Questionnaire cohort | | | | |
| Norway, Bergen | ECRHS I quest. respondents | 1250 | National registers | 1763 |
| Denmark, Aarhus | ECRHS I quest. respondents | 974 | National registers | 1224 |
| Sweden, Uppsala | RHINE III quest. respondents | 894 | National registers | 1314 |
| Sweden, Göteborg | RHINE III quest. respondents | 709 | National registers | 951 |
| Sweden, Umeå | RHINE III quest. respondents | 876 | National registers | 1307 |
| Iceland, Reykjavik | ECRHS I quest. respondents | 977 | National registers | 1245 |
| Estonia, Tartu | ECRHS I quest. respondents | 525 | National registers | 618 |
| Australia, Melbourne | ECRHS III clin. respondents | 149 | Through the parents | 245* |
| Spain, Huelva | ECRHS III clin. respondents | 48 | Through the parents | 72* |
| Spain, Albacete | ECRHS III clin. respondents | 39 | Through the parents | 79* |
| Clinical cohort | | | | |
| Norway, Bergen | ECRHS III clin. respondents | 346 | National registers | 499 |
| Denmark, Aarhus | ECRHS III clin. respondents | 68 | National registers | 83 |
| Sweden, Uppsala | ECRHS III clin. respondents | 74 | National registers | 90 |
| Sweden, Göteborg | ECRHS III clin. respondents | 53 | National registers | 60 |
| Sweden, Umeå | ECRHS III clin. respondents | 66 | National registers | 86 |
| Iceland, Reykjavik | ECRHS III clin. respondents | 97 | National registers | 120 |
| Estonia, Tartu | ECRHS III clin. respondents | 159 | National registers | 195 |
| Australia, Melbourne | ECRHS III clin. respondents | 102 | Through the parents | 144 |
| Spain, Huelva | ECRHS III clin. respondents | 38 | Through the parents | 62 |
| Spain, Albacete | ECRHS III clin. respondents | 38 | Through the parents | 66 |

*Parental (G2) information extracted from the ECRHS and harmonised with RHINE questions.
ECRHS, European Community Respiratory Heath Survey; RHINE, Respiratory Health In Northern Europe; RHINESSA, Respiratory Health in Northern Europe, Spain and Australia.

medication). Study centres in Estonia with recent transition from middle- to high-income economy, Spain as a southern European country, and Australia with particularly high allergy prevalence, extend the generalisability of study results beyond Northern Europe where most study centres are situated.

### Offspring cohort (G3)
The RHINESSA adult offspring study invited offspring age ≥18 years (G3) of RHINE and ECRHS participants (G2) from ten study centres: Bergen, Norway; Aarhus, Denmark; Uppsala, Göteborg and Umeå, Sweden; Reykjavik, Iceland; Tartu, Estonia; Melbourne, Australia; Huelva and Albacete, Spain (table 1, figure 2). In the Northern European countries all G3 offspring were identified through national registries, for the Spanish and Australian study centres the G3 offspring's contact details were obtained from the G2 parents in ECRHS III (table 1). All offspring with parental questionnaire information (from RHINE or ECRHS) were invited to a questionnaire study. The subsample of these with parental clinical information (from ECRHS) and residing in the

study area, were invited to a clinical study (online supplemental figure S1). The baseline data collection was performed in all study centres during 2014–2017. The same study protocols (adapted to age) were used in all study centres and all generations, and detailed standard operating procedures (see www.rhinessa.net), interview guides and procedures for translations/back translations contribute to secure harmonisation of data across study centres and generations.

The study centres in Bergen, Aarhus, Tartu and Melbourne also investigated younger G3 offspring <18 years (online supplemental table S1). Bergen RHINESSA further investigated the G1 grandparent generation, the G2 parent not participating in RHINE/ECRHS, and the G4 offspring's offspring. Summary data for these additional cohorts are given in the online supplemental table S2.

### Parental cohorts (G2)
In the early 1990's the ECRHS conducted a population-based survey among random samples of young adults aged 20–44 years in several European and non-European

countries (www.ecrhs.org).[16] On average 4000 persons (range 1000–7000) from each centre were invited to a postal survey (mean response rate 73%). Clinical examinations were conducted in subsamples from ~45 study centres, primary random subsamples, but for some centres an additional subsample with persons with asthma symptoms. ECRHS followed up the clinical samples in 30 study centres in 2002–2004 (ECRHS II)[17] and 2012–2015 (ECRHS III).[18]

The RHINE study (www.rhine.nu) developed protocols to follow-up responders to the initial ECRHS postal survey in seven Northern European centres: Bergen, Norway; Göteborg, Umeå and Uppsala, Sweden; Aarhus, Denmark; Reykjavik, Iceland and Tartu, Estonia. In 2000–2002, 16 106 persons answered extensive postal questionnaires (RHINE II, mean response rate 75%).[19 20] The population was reinvestigated in 2010–12, with 13 093 answering a postal questionnaire. Analyses of non-response showed only minor differences between long-term participants and baseline participants in exposure-outcome associations between age, gender, smoking and respiratory symptoms.[19]

## Follow-up

Regular follow-up of the RHINESSA clinical and questionnaire cohorts is planned to take place with 5–8 years intervals. The first follow-up of the full cohort is about to start in all study centres in 2021–2022. An ad hoc clinical follow-up was performed in Bergen study centre in 2020 to capture features related to the COVID-19 pandemic. The parent populations of all study centres have been followed with 8–10 years intervals since the 1990s, and the fourth study waves of RHINE and ECRHS are taking place in 2021–2022.

## Ethical consideration

Ethical permissions were obtained for each study wave from the local ethics committee in each of the participating centres. The ethical approval reference numbers are listed on www.rhinessa.net. All study participants provided written informed consent prior to participation. Permission to extract information about themselves and family members from national registers were obtained from each participant in the Northern European study centres. For children and adolescents participating in the additional study groups presented in the online supplemental file, written informed consents were given by the parents/guardian, as required by the local ethics committees.

## Response rate and parental characteristics related to offspring response

Identified offspring were sent an invitation letter with information about the study and log-in details to a web-based questionnaire, two reminders were sent, in some centres including a postal questionnaire. Persons eligible to a clinical study, were invited by a letter and/or contacted by telephone to agree on an appointment for clinical investigation, also with two reminders. For the three Swedish study centres, the researchers were not allowed to identify and approach study participants directly and Statistics Sweden distributed the invitation letters to participants of both the questionnaire and the clinical study. Altogether 8818 persons participated in the questionnaire cohort and 1405 of these in the clinical cohort (table 1). The overall response rate was 35% both for the questionnaire and the clinical cohort, with differences between study centres and between the questionnaire and clinical stages (table 2). Reasons for non-participation included inability to make contact with the persons due to erroneous contact details or because the person was no longer living at

**Table 2** Response rate for RHINESSA adult offspring (18+ years) participants (G3) for the questionnaire cohort and the clinical cohort eligible subjects were defined as live subjects 18 years and older with known residential addresses residing in the country (questionnaire cohort) or in/near the study centre (clinical cohort)

| | Questionnaire cohort (G3) | | | Clinical cohort (G3) | | |
|---|---|---|---|---|---|---|
| Centre | Eligible, N | Included, N | Response rate, % | Eligible, N | Included, N | Response rate, % |
| Bergen | 4385 | 1763 | 40.2 | 1278 | 499 | 39.0 |
| Aarhus | 4014 | 1224 | 30.5 | 381 | 83 | 21.8 |
| Uppsala, Göteborg, Umeå | 8256 | 3572 | 42.7 | 639 | 236 | 36.9 |
| Reykjavik | 4756 | 1245 | 26.2 | 200 | 120 | 60.0 |
| Tartu | 2907 | 618 | 21.3 | 669 | 195 | 29.1 |
| Melbourne | 499 | 245 | 49.1 | 245 | 144 | 58.8 |
| Huelva | 244 | 72 | 29.5 | 244 | 66 | 27.0 |
| Albacete | 365 | 79 | 21.6 | 365 | 62 | 17.0 |
| Total | 25 426 | 8818 | 34.7 | 4021 | 1405 | 34.9 |

RHINESSA, Respiratory Health in Northern Europe, Spain and Australia.

**Table 3** Parental (G2) characteristics for RHINESSA (G3) adult (18+ years) responders compared with the source parental RHINE/ECRHS population (G2) for the questionnaire cohort and the clinical cohort

| | Questionnaire cohort | | Clinical cohort | |
|---|---|---|---|---|
| | RHINE/ECRHS parents (G2) to RHINESSA adult offspring (G3) N=6441 | All RHINE (G2) N=13 260 | RHINE/ECRHS parents (G2) to RHINESSA adult offspring (G3) N=1041 | All* ECRHS (G2) N=3205 |
| Paternal (G2) characteristics | | | | |
| Ever smoker, % | 53.2 | 54.6 | 55.9 | 55.3 |
| Primary school | 17.1 | 11.9 | 14.2 | 12.1 |
| Secondary school | 37.1 | 44.3 | 38.1 | 45.4 |
| College/University | 45.8 | 43.8 | 47.7 | 42.5 |
| BMI (SD) | 26.8 (4) | 26.8 (4) | 27.8 (4) | 27.6 (4) |
| Overweight in puberty,† % | 9.2 | 10.3 | 9.9 | 10.9 |
| Asthma, % | 12.3 | 10.3 | 23.7 | 17.6 |
| Wheeze, % | 20.1 | 20.7 | 28.6 | 27.1 |
| Maternal (G2) characteristics | | | | |
| Ever smoker, % | 54.0 | 54.0 | 45.1 | 52.1 |
| Primary school | 16.2 | 10.9 | 16.4 | 12.3 |
| Secondary school | 33.9 | 39.5 | 33.9 | 38.2 |
| College/University | 49.9 | 49.6 | 49.7 | 49.5 |
| BMI (SD) | 25.7 (5) | 25.6 (5) | 27.0 (5) | 27.0 (5) |
| Overweight in puberty,† % | 23.9 | 24.3 | 23.5 | 23.5 |
| Asthma, % | 14.3 | 13.6 | 23.1 | 26.4 |
| Wheeze, % | 20.2 | 19.3 | 24.4 | 27.7 |

*Only including data for the 10 study centres in RHINESSA.
†Overweight defined by self-reported body silhouette at age of menarche/age of voice break.[29]
BMI, body mass index; ECRHS, European Community Respiratory Heath Survey; RHINE, Respiratory Health In Northern Europe; RHINESSA, Respiratory Health in Northern Europe, Spain and Australia.

that address, as well as unwillingness or inability to participate. However, parental characteristics were compared between the responders and the source parental population (table 3), showing fairly similar characteristics and no clear patterns of differences, for example, approximately 55% had a father or mother who had ever smoked in both groups, and there was, for example, no clear trend of higher response rates among offspring of symptomatic parents. As expected due to the original sampling strategy in ECRHS (enriched with persons with symptoms) the prevalence of asthma is somewhat higher in the clinical sample compared with the questionnaire sample.

### Collected data and characteristics of study participants
Data and biomaterial collected in RHINESSA include questionnaire/interview information on respiratory and general health, life style and environmental exposures; measurements of lung function, anthropometrics, blood pressure; allergy markers, sex hormones, DNA methylation, and biomarkers in urine and dust samples. Table 4 displays questionnaire/interview data, clinical measures, samples, and measured

biomarkers that is available in the RHINESSA adult (18+ years) population (G3), as well as information that has been collected from/about their parents (G2), their grandparents (G4) and their offspring (G1). In addition, national health registries in the Nordic countries with excellent coverage provide an additional data source for the generations G1–G4 and their family members. Some registries date back to the 18th century, while there are most registry data available the last decades.

Table 5A,B displays characteristics of the study population by study centre, separately presented for the questionnaire study population (5a) and the clinically investigated subsample (5b). Mean age at baseline was 30.1 years and there were 58% women, 33% had ever smoked and 21% had ever used oral moist tobacco (0.8% in Aarhus, 33.9% in Umeå). Asthma medication was used at the time of study by 8.7%, ranging from 6.5% in Aarhus to18.9% in Melbourne (table 5). The proportion of missing data ranges from <0.01% to 4.2% for key variables presented in table 5A,B.

Definitions of asthma, hay fever/nose allergies and atopic dermatitis are the same as in the questionnaire

**Table 4** Key data available for the G3 RHINESSA adult offspring (18+ years), and their G1 grandparents, G2 parents and G4 offspring. For the G2 generation, information from three study waves are presented (x available for all; ss available in subsample– see footnotes)

| | Grandparents (G1) born 1898–1965 | Parents (G2) at 20–44 years RHINE/ ECRHS I | Parents (G2) at 30–54 years RHINE/ ECRHS II | Parents (G2) at 40–64 years RHINE/ ECRHS III | RHINESSA (G3) adult offspring 18–53 years* | Offspring's (G4) offspring age 0–30 years |
|---|---|---|---|---|---|---|
| Questionnaire/interview | | | | | | |
| Social class, education | x | x | x | x | x | ss |
| Childhood factors | x | x | x | x | x | ss |
| Birth characteristics† | | ss | | | | ss |
| Place of upbringing | x | x | x | x | x | ss |
| Smoking | x | x | x | x | x | ss |
| Snus, oral moist tobacco, E-cigarettes | | | | | x | ss |
| Occupation | ss | x | x | x | x | ss |
| Indoor environment | ss | x | x | x | x | ss |
| Personal care products | ss | | | x | x | ss |
| Height/weight | ss | x | x | x | x | ss |
| Body shapes | x | | | x | x | ss |
| Waist circumference (self-measured) | | | | x | x | |
| Physical activity | ss | ss | x | x | x | ss |
| Diet | ss | ss | ss | x | x | ss |
| Allergic diseases/ symptoms | x | x | x | x | x | x |
| Asthma/symptoms | x | x | x | x | x | x |
| Asthma/allergy treatment | ss | x | x | x | x | ss |
| Sleep | ss | ss | ss | x | x | ss |
| Other diseases/ symptoms | x | x | x | x | x | ss |
| Quality of life SF-36/RAND | ss | | ss | ss | ss | ss |
| Work disability | | | ss | x | x | |
| Air pollution and greenness‡ | | | | ss | ss | |
| Women questionnaire/interview (from women in each cohort) | | | | | | |
| Pregnancies and complications | | | x | x | x | ss |
| Birth characteristics of offspring | | | x | x | x | ss |
| Menarche, menstrual data, menopause | | | x | x | x | ss |
| Exogenous sex hormones | | | x | x | x | ss |

Continued

**Table 4** Continued

| | Grandparents (G1) born 1898–1965 | Parents (G2) at 20–44 years RHINE/ECRHS I | Parents (G2) at 30–54 years RHINE/ECRHS II | Parents (G2) at 40–64 years RHINE/ECRHS III | RHINESSA (G3) adult offspring 18–53 years* | Offspring's (G4) offspring age 0–30 years |
|---|---|---|---|---|---|---|
| Irregular menstruation, PCOS | | x | x | x | | ss |
| Gynaecological and related diseases | | x | x | x | | ss |
| **Clinical measures (from clinical stage in each cohort)** | | | | | | |
| Anthropometry (height/weight/waist/hip) | ss | x | x | x | x | ss |
| Bioimpedance | ss | | | x | ss | ss |
| Spirometry (FEV1, FVC) | ss | x | x | x | x | ss |
| Post-BD spirometry | ss | | | x | x | ss |
| Metacholine test | | x | x | | | |
| FeNO | ss | | | x | x | ss |
| Skin prick test | ss | x | | x | x | ss |
| Blood pressure | ss | | | x | x | ss |
| Heart rate | ss | | | x | x | ss |
| CIMT (carotis intima media) | | | | | ss | |
| CPI/caries index | | | | ss | ss | |
| **Biological material and environmental samples (from clinical stage in each cohort)** | | | | | | |
| Blood samples | ss | x | x | x | x | ss |
| Gingival samples | ss | | | ss | ss | ss |
| Skin swab | ss | | | | ss | ss |
| Saliva | ss | | | | ss | ss |
| Urine | ss | ss | | x | x | ss |
| Bedroom dust samples | ss | | ss | ss | ss | ss |
| **Biomarkers measured/funded at time of publication (from clinical stage in each cohort)** | | | | | | |
| Total and specific IgEs | ss | x | x | x | x | ss |
| Genome wide genotyping | | ss | ss | | | |
| Selective genotyping | | ss | ss | | | |
| DNA methylation in fullblood | ss | ss | ss | ss | ss | |
| Fasting blood glucose | ss | | | ss | ss | ss |
| Sex hormones | | women | women | women | ss | ss |
| Oral microbiome | ss | | | | ss | |
| Urine biomarkers of chemical exposures | | | | | ss | |

Continued

**Table 4** Continued

| | Grandparents (G1) born 1898–1965 | Parents (G2) at 20–44 years RHINE/ECRHS I | Parents (G2) at 30–54 years RHINE/ECRHS II | Parents (G2) at 40–64 years RHINE/ECRHS III | RHINESSA (G3) adult offspring 18–53 years* | Offspring's (G4) offspring age 0–30 years |
|---|---|---|---|---|---|---|
| Complete blood cell counts | | | | | ss | |
| Adipokines | | ss | ss | | | |

Subsamples.

Grandparents and offsprings' offspring were only investigated in Bergen, information in other centres are given by family members.

CIMT and CPI were only measured in Bergen.

Gingival samples were collected in parents and offspring from Bergen, Melbourne and Tartu, and in offspring from Uppsala.

DNA methylation was measured in fullblood using the Illumina EPIC BeadChip arrays in approximately 900 offspring, 400 parents and 140 grandparents.

Sex hormones were measured in mothers and approx. 1000 offspring from all centres.

Oral microbiome was measured using 16S rRNA Illumina MiSeq in Bergen adult/adolescent offspring and grandparent.

Urine biomarker concentrations of chemical exposure was measured in Bergen adult/adolescent offspring.

Complete blood cell counts were measured in Swedish centres, adipokines also in Reykjavik.

Helminth serology was measured in offspring from Bergen, Tartu and Aarhus, and parents from Bergen.

*Sweden and Iceland did a shorter clinical protocol of RHINESSA adult offspring, not including bioimpedance, skin swap or saliva (except that Uppsala collected saliva). RHINESSA offspring <18 years were included in Aarhus, Bergen, Melbourne and Tartu, following age-adapted slightly shorter protocols, similar to protocols used for corresponding age groups in offspring's offspring.

†Information from registries and hospital protocols, and from mothers.

‡Information generated using geocoding based on registry data on life-time addresses.

BD, bronchodilator; CPI, Community Periodontal Index; ECRHS, European Community Respiratory Heath Survey; FeNO, Fractional exhaled nitric oxide; FEV1, Forced Expiratory Volume in 1 second; FVC, Forced Vital Capacity; PCOS, Polycystic ovary syndrome; RHINE, Respiratory Health In Northern Europe; RHINESSA, Respiratory Health in Northern Europe, Spain and Australia; SF-36, Short Form 36 Health Survey Questionnaire.

study presented above in table 2, but based on information obtained by standardised interviews rather than self-filled in questionnaires.

The presented lung function data refer to prebronchodilator measurements.

## Participant and public involvement

User representatives from the Norwegian Asthma and Allergy Foundation and the Norwegian Labour Inspection have been involved in the RHINESSA Advisory Board from the establishment of the study, and have contributed to development of the study as well as discussions of priorities in analyses and publication of data. One study participant has at a later stage been included in the RHINESSA Advisory Board, as user representative of the study population. Information on the research is available to the study participants through the study website and newsletters. Field workers are alert to comments from the study participants regarding the burden of the study and convey these experiences in annual meetings.

## FINDINGS TO DATE

A summary of key findings to date is provided in table 6.

## Smoking and overweight in male prepuberty and offspring health

An explorative analysis of asthma in >24 000 offspring of the RHINE cohort,[21] suggested that father's smoking before conception was associated with asthma in future offspring. Mother's smoking around the time of pregnancy, but not before conception, and paternal grandmother's smoking were further associated with offspring asthma. A multigeneration analysis of the ECRHS cohort[22] by Accordini et al confirmed effects of father's pre-pubertal smoking, using advanced statistical mediation modelling to account for the complexity in the multicentre multigeneration data, including simulation analyses showing that the impact of unmeasured confounding on the estimates was limited.[23 24] State-of-the-art statistical methods for causal inference from observational data[25] were applied in a subsequent analysis of the RHINESSA/ECRHS cohorts, suggesting that father's smoking <15 years caused lower lung function in offspring.[26] Effects of father's smoking across generations are supported by preliminary mechanistic work, including a study by Mørkve Knudsen et al showing that father's smoking was associated with specific DNA methylation patterns in adult offspring,[27] and a murine study by Hammer et al uncovering that preconception smoke exposure altered miRNAs in the spermatozoa, and gave higher postnatal body weight in progeny.[28]

Further support for early male puberty as an important susceptibility window, was revealed by Johannessen et al showing that father's onset of overweight between age 8 years and voice break was associated with asthma in future offspring.[29] An ongoing analysis by Lønnebotn et al suggests that father's prepubertal overweight also may cause lower lung function in offspring.[30] Investigating overweight as

**Table 5** Characteristics of the RHINESSA adult offspring (18+ years) cohorts by centre; (A) questionnaire cohort (N=8818), and (B) clinical cohort (N=1405)

| | Bergen | Aarhus | Uppsala | Göteborg | Umeå | Reykjavik | Tartu | Melbourne | Huelva | Albacete | Total | Missing, % |
|---|---|---|---|---|---|---|---|---|---|---|---|---|
| **(A)** | | | | | | | | | | | | |
| Age (mean, SD) | 29.2 (7.4) | 27.0 (7.4) | 30.4 (7.6) | 31.5 (8.0) | 32.0 (7.5) | 32.0 (8.1) | 28.6 (6.2) | 28.9 (6.5) | 32.5 (7.0) | 30.6 (7.1) | 30.1 (7.7) | 0.4 |
| Sex, % females | 57.8 | 59.8 | 56.3 | 52.9 | 57.0 | 62.7 | 58.4 | 53.7 | 62.5 | 52.6 | 57.8 | <0.1 |
| BMI (mean, SD) | 24.3 (4.3) | 23.7 (4.3) | 24.0 (4.2) | 24.5 (4.3) | 24.6 (4.4) | 26.2 (5.1) | 23.8 (4.5) | 23.7 (4.8) | 24.2 (4.1) | 23.9 (5.2) | 24.4 (4.5) | 3.2 |
| Ever smoker, % | 36.5 | 30.3 | 29.5 | 35.1 | 26.0 | 38.3 | 38.0 | 31.2 | 41.7 | 55.1 | 33.3 | 1.9 |
| Ever used oral moist tobacco,* % | 29.6 | 4.1 | 23.1 | 24.3 | 33.9 | 15.7 | 9.2 | 0.8 | N/A | N/A | 20.5 | 0.5 |
| Current smoker, % | 12.7 | 15.2 | 8.8 | 14.4 | 7.6 | 14.1 | 21.5 | 13.9 | 33.3 | 32.1 | 13.0 | 2.0 |
| Domestic ETS in childhood, % | 54.8 | 50.8 | 37.0 | 49.3 | 43.5 | 61.2 | 55.8 | 24.6 | 54.2 | 63.6 | 49.4 | 3.7 |
| Educational level | | | | | | | | | | | | |
| Primary school, % | 2.6 | 2.1 | 2.5 | 2.4 | 2.2 | 5.3 | 7.2 | 0.0 | 1.4 | 6.6 | 3.1 | 2.0 |
| Secondary educ. % | 35.9 | 43.3 | 37.3 | 45.3 | 42.4 | 33.1 | 38.2 | 22.2 | 40.3 | 32.9 | 38.5 | |
| College, univ. % | 61.5 | 54.6 | 60.2 | 52.3 | 55.4 | 61.6 | 54.6 | 77.8 | 58.3 | 60.5 | 58.4 | |
| Childhood asthma (onset <10 years), % | 6.3 | 5.9 | 6.4 | 5.0 | 7.9 | 10.8 | 3.6 | 25.0 | 13.9 | 3.9 | 7.4 | 1.1 |
| Current asthma medication, % | 7.6 | 6.5 | 9.9 | 8.1 | 11.4 | 8.6 | 3.9 | 18.9 | 9.7 | 14.3 | 8.7 | <0.1 |
| Current hay fever/nose allergy, % | 28.9 | 25.7 | 29.6 | 27.9 | 26.9 | 32.2 | 27.3 | 47.3 | 36.1 | 35.1 | 29.1 | 0.4 |
| Childhood atopic dermatitis (onset <10 years), % | 6.7 | 8.5 | 8.9 | 9.7 | 8.0 | 10.3 | 5.3 | 9.8 | 0.0 | 3.9 | 8.2 | 4.0 |
| Current atopic dermatitis, % | 8.6 | 8.2 | 13.0 | 11.3 | 10.0 | 14.5 | 11.0 | 11.1 | 5.6 | 12.8 | 10.8 | <0.1 |
| **(B)** | | | | | | | | | | | | |
| Age (SD) | 28.0 (6.6) | 28.2 (8.2) | 31.4 (7.8) | 31.3 (7.5) | 31.2 (7.3) | 34.6 (8.1) | 29.8 (5.8) | 29.1 (6.5) | 32.2 (7.1) | 31.0 (7.7) | 29.9 (7.2) | 0.2 |
| Sex, % females | 47.3 | 62.2 | 68.9 | 55.2 | 55.3 | 56.8 | 44.3 | 53.2 | 63.6 | 47.5 | 52.1 | 0.2 |
| BMI (SD) | 25.1 (4.5) | 24.4 (5.0) | 25.3 (5.5) | 24.6 (3.5) | 25.2 (4.4) | 28.1 (5.1) | 24.9 (5.0) | 25.0 (4.5) | 24.3 (4.5) | 24.2 (5.0) | 25.2 (4.8) | 0.3 |
| Ever smoker, % | 29.9 | 25.3 | 30.0 | 30.0 | 17.4 | 45.0 | 38.5 | 26.4 | 43.9 | 51.6 | 32.6 | 0.6 |
| Current smoker, % | 13.9 | 16.9 | 4.5 | 5.0 | 1.2 | 9.2 | 19.5 | 6.9 | 28.8 | 29.0 | 13.4 | 0.2 |

Continued

**Table 5** Continued

| | Bergen | Aarhus | Uppsala | Göteborg | Umeå | Reykjavik | Tartu | Melbourne | Huelva | Albacete | Total | Missing, % |
|---|---|---|---|---|---|---|---|---|---|---|---|---|
| Domestic ETS in childhood,† % | 54.2 | 44.6 | N/A | N/A | N/A | N/A | 53.9 | 36.8 | 53.0 | 75.8 | 52.2 | *0.8* |
| Childhood asthma, onset <10 years, % | 4.4 | 8.4 | 6.7 | 3.5 | 5.8 | 10.9 | 2.1 | 19.4 | 9.2 | 3.2 | 6.8 | *1.1* |
| Current asthma medication, % | 4.6 | 3.6 | 8.9 | 0.0 | 5.8 | 2.5 | 1.5 | 18.1 | 6.2 | 6.5 | 5.6 | *3.1* |
| Current hay fever/nose allergy, % | 31.5 | 31.3 | 38.2 | 50.0 | 37.7 | 29.2 | 26.7 | 51.4 | 31.8 | 29.0 | 34.1 | *3.7* |
| Childhood atopic dermatitis, onset <10 years),‡ % | 5.4 | 2.4 | 5.6 | 3.3 | 4.7 | 9.2 | 5.1 | 20.1 | 3.0 | 1.6 | 6.6 | *0.9* |
| Current atopic dermatitis, % | 7.2 | 6.0 | 10.1 | 10.0 | 5.8 | 14.3 | 3.6 | 16.0 | 6.1 | 3.2 | 8.1 | *0.1* |
| $FEV_1$ l (SD) | 3.91 (0.8) | 3.80 (0.7) | 3.56 (0.7) | 3.77 (0.7) | 3.72 (0.7) | 3.64 (0.8) | 4.08 (0.8) | 3.78 (0.8) | 3.51 (0.7) | 3.63 (0.6) | 3.83 (0.8) | *1.6* |
| FVC l (SD) | 4.80 (1.1) | 4.65 (0.9) | 4.42 (0.9) | 4.70 (0.9) | 4.73 (0.9) | 4.60 (1.0) | 4.97 (1.0) | 4.69 (0.9) | 4.29 (0.9) | 4.28 (0.8) | 4.73 (1.0) | *1.9* |
| $FEV_1$/FVC (SD) | 0.82 (0.1) | 0.82 (0.1) | 0.81 (0.1) | 0.80 (0.1) | 0.79 (0.1) | 0.79 (0.1) | 0.82 (0.1) | 0.81 (0.1) | 0.82 (0.1) | 0.85 (0.1) | 0.81 (0.1) | *2.0* |
| $FEV_1$ % pred.† (SD) | 95 (11) | 94 (10) | 94 (13) | 93 (11) | 93 (10) | 94 (12) | 97 (11) | 95 (13) | 96 (11) | 95 (11) | 95 (11) | *4.2* |
| FVC % pred.§ (SD) | 98 (11) | 97 (10) | 97 (12) | 96 (10) | 98 (10) | 98 (11) | 98 (10) | 99 (11) | 97 (10) | 93 (12) | 98 (11) | *4.2* |
| $FEV_1$/FVC% pred (SD) | 97 (7) | 96 (6) | 96 (7) | 96 (6) | 94 (6) | 96 (6) | 98 (7) | 95 (7) | 98 (7) | 102 (7) | 97 (7) | *4.2* |

Numbers in italic refer to percentage of missing values for that variable in the total group.

*Not available information on e-cigarettes in centres labelled N/A.

†Not available information on ETS exposure in centres labelled N/A.

‡Defined as ever having had itchy rash that was coming and going for at least 6 months, and that the rash affected any of the following places: the fold of the elbows, behind the knees, in front of the ankles, under the buttocks or around the neck, ears or eyes.

§Calculated based on Global Lung function Initiative GLI2012 reference values (Quanjer et al, ERJ 2012),[47]

BMI, body mass index; ETS, environmental tobacco smoke; FEV1, Forced Expiratory Volume in 1 second; FVC, Forced Vital Capacity; N/A, not applicable; RHINESSA, Respiratory Health in Northern Europe, Spain and Australia.

**Table 6** Overview of key publications from RHINESSA/RHINE/ECRHS on preconception exposures as related to offspring respiratory outcomes, phenotypes across generations and validation studies relevant for multigeneration research

| Exposure | Outcome | Exposure window | Main findings | Study cohorts | Reference |
|---|---|---|---|---|---|
| **Smoking** | | | | | |
| Smoking | Non-allergic early onset asthma | Paternal prepuberty: paternal grandmother's pregnancy | Fathers smoking in prepuberty associated with asthma in his offspring, in absence of grandmothers smoking during the father's pregnancy. | RHINE | Svanes et al[21] |
| Smoking | Allergic and non-allergic asthma | Paternal prepuberty; pregnancy | Fathers smoking in prepuberty associated with non-allergic asthma in his offspring; grandmothers smoking during mother's fetal period associated with allergic asthma in her grandchild. | ECRHS | Accordini et al[22] |
| Smoking | Lung function | Paternal prepuberty; grand-maternal pregnancy | Fathers smoking in prepuberty reduced offspring's $FEV_1$, and FVC; the grandmothers smoking during the father's fetal period reduced the grandchild's $FEV_1$/FVC. | Parents: ECRHS Offspring: RHINESSA | Accordini et al[26] |
| **Occupational exposures** | | | | | |
| Welding | Non-allergic asthma | Paternal adolescence | Fathers' preconception welding was associated with non-allergic asthma in offspring. | RHINE | Svanes et al[21] |
| Allergens, reactive chemicals, micro-organisms and pesticides | Asthma | Before conception of child; preconception and postconception combined | Preconception maternal and paternal exposure to occupational agents not associated with asthma in offspring, expect higher early-onset asthma with mother exposure to allergens and/or reactive chemicals before and after conception | Parents: ECRHS Offspring: RHINESSA | Pape et al[36] |

**Table 6** Continued

| Exposure | Outcome | Exposure window | Main findings | Study cohorts | Reference |
|---|---|---|---|---|---|
| Cleaning products and disinfectants | Asthma and/or wheeze | Before conception of child; around conception and pregnancy | Mother's exposure to indoor cleaning starting before conception was associated with offspring's childhood allergic and non-allergic asthma. | Parents: RHINE Offspring: RHINESSA | Tjalvin et al[37] |
| Environmental exposures | | | | | |
| Air pollution | Asthma and allergies | Parental childhood | Parental exposure to air pollution during childhood increased the risk of asthma and allergies in offspring. | RHINESSA | Kuiper et al[38] |
| Farm exposure | Asthma | Parental childhood | Farm upbringing in previous generations was not associated with offspring asthma—either for parental or grandparental upbringing. | Parents: ECRHS/ RHINE Offspring: RHINESSA | Timm et al[34] |
| Metabolic and hormonal exposures | | | | | |
| Overweight and weight gain | Non-allergic asthma | Paternal puberty | Paternal overweight and weight gain before puberty associated with offspring non-allergic asthma. | Parents: ECRHS/ RHINE Offspring: RHINESSA | Johannessen et al[29] |
| Overweight | Lung function | Paternal childhood/puberty | Paternal overweight during childhood and/or puberty may cause lower lung function in offspring. | Parents: ECRHS Offspring: RHINESSA | Lønnebotn et al[30] |
| Infections and disease processes | | | | | |
| Helminth infection | Allergies | Not known | *Toxocara spp* seropositivity in parents was associated with allergic outcomes in their offspring. | Parents: ECRHS Offspring: RHINESSA | Jogi et al[33] |
| Bronchial hyper-responsiveness and level of specific IgEs | Asthma and allergies | Before conception of child | Parental asthmatic and allergic disease activity measured before conception was associated to offspring asthma and hay fever. | ECRHS | Bertelsen et al[32] |
| Phenotype across generations | | | | | |

Continued

**Table 6** Continued

| Exposure | Outcome | Exposure window | Main findings | Study cohorts | Reference |
|---|---|---|---|---|---|
| Sleep characteristics | | | Sleep-related symptoms and sleep duration more common in offspring with same outcome in parents, after adjusting for lifestyle factors, education and parity in both generations | Parents: ECRHS/RHINE Offspring: RHINESSA | Lindberg et al[39] |
| Breathlessness | | | Breathlessness nearly doubled in offspring of parents with breathlessness, after adjusting for obesity, smoking, depression, asthma, lower lung function and female sex in both generations | Parents: ECRHS/RHINE Offspring: RHINESSA | Ekstrøm et al[40] |
| Validation studies | | | | | |
| Asthma reported by family members | | | Moderate to good agreement between self-reported asthma and asthma reported by family members, for offspring asthma reported by parents and vice versa, better fr childhood than adult onset asthma. | Parents: ECRHS/RHINE Offspring: RHINESSA | Kuiper et al[41] |
| Parental smoking reported by offspring | | | Adults reported well their parents' smoking during their childhood and their mother' smoking when pregnant with them, when compared with the parents' own report. | Parents: ECRHS/RHINE Offspring: RHINESSA | Pape et al[42] |
| Parents' place of upbringing reported by offspring | | | Offspring report of parents' place of upbringing was dependent on own place of upbringing, this did not bias the associations of place of upbringing with asthma.[43] | Parents: ECRHS/RHINE Offspring: RHINESSA | Timm et al[43] |
| Birth characteristics reported by mothers | | | High validity for mother's report of birth and pregnancy parameters. Risk-associations were similar when using maternal vs registry-based information. | Bergen RHINE, Medical Birth Registry of Norway | Skulstad et al[44] |
| Current body silhouettes validated against measured and reported height/weight | | | Current body silhouettes were highly correlated with BMI calculated from either measured or self-reported weight and height. | ECRHS, RHINE | Dratva et al[45] |
| Retrospective body silhouettes validated against previously measured and reported height/ weight | | | Retrospective body silhouettes from adult ages correlated well with BMI calculated from measured height/weight at corresponding ages in the past, and allowed differentiation of obesity and non-obesity | ECRHS, RHINE | Lønnebotn et al[46] |

BMI, body mass index; ECRHS, European Community Respiratory Heath Survey; FEV1, Forced Expiratory Volume in 1 second; FVC, Forced Vital Capacity; RHINE, Respiratory Health In Northern Europe; RHINESSA, Respiratory Health in Northern Europe, Spain and Australia.

outcome, Knudsen *et al* demonstrated that father's prepubertal smoking onset was associated with excessive fat mass in their future sons.[31] Johannessen *et al* showed that father's and mother's overweight in childhood, and mother's overweight at menarche, were associated with offspring overweight in childhood.[29]

## Other preconception exposures in mothers and fathers and offspring health

Bertelsen *et al* found that parental asthmatic and allergic disease activity *before* conception was more strongly associated with offspring allergic asthma, than parental disease activity after the child was born.[32] The identified pattern might possibly reflect an influence of asthmatic/allergic disease activity on germline cells and thereby on future offspring phenotype. A study of parental immune response to helminths in Norway by Jõgi *et al* uncovered that IgG4 to the zoonotic helminth *Toxocara* in parents was associated with allergic symptoms in their offspring, following a sex-specific pattern.[33] Timm *et al* explored whether farm upbringing in previous generations could influence offspring asthma and allergies, and found no evidence of an association between parental/grandparental farm upbringing and offspring asthma.[34] Regarding selective migration which has not previously been studied across three generations, an analysis suggested that asthma in the family was not a risk factor for quitting farming.[35] Regarding parental occupational exposures, Svanes *et al* found that father's welding ≥10 years before conception was associated with a doubled risk of asthma in future offspring.[21] Pape *et al* investigated four groups of exposures defined from an asthma-specific job exposure matrix, and compared exposure only before conception with exposure starting before conception and continuing. Associations with offspring asthma were not identified for most exposure groups, except higher risk of early-onset asthma for mothers' exposure to 'allergens and reactive chemicals' before and after the offspring's birth.[36] Tjalvin *et al* investigated the specific exposure category 'indoor cleaning agents: cleaning products/detergents and disinfectants', present in jobs codes such as nurses, personal care workers, cooks and cleaner.[37] Exposure starting before conception was associated with higher asthma risk in offspring, while there was no association with exposure starting after birth. Kuiper *et al*[38] analysed parental air pollution *in* childhood/adolescence as related to offspring asthma and hayfever. Data on various air pollutants in parents from 1975 onwards were generated by geocoding of parental individual residential addresses obtained from national registries. Mother's PM10 exposure before age 18 years had a direct effect with doubled asthma risk in offspring, and father's ozone exposure in the same age window was associated with increased offspring hayfever risk.[38]

## Heritability in symptoms and diseases across generations

In a study of sleep disturbances, Lindberg *et al* showed that sleep-related symptoms and sleep duration were more common in offspring whose parents had reported the same symptom, consistent after adjusting for lifestyle factors, education and parity.[39] Ekström *et al* found that breathlessness was nearly doubled in offspring of parents with breathlessness, even when adjusting for factors associated with breathlessness in both generations (obesity, smoking, depression, asthma, lower lung function and female sex).[40] Carsin *et al* found that grandfather's cardiometabolic disease (CMD) was directly associated with grand offspring asthma, while accounting for indirect effects transmitted through parental CMD or asthma, consistently in the RHINE, ECRHS and RHINESSA cohorts (not yet published).

## Validation studies informing multigeneration epidemiological research

Information about family members is often sought from study participants, as this is cost-effective and may be the only feasible way to obtain the information. The RHINESSA/RHINE/ECRHS cohorts provide an important opportunity for validation of such next of kin information. Kuiper *et al* found moderate to good agreement between self-reported asthma and asthma reported by family members, both regarding offspring asthma reported by parents and vice versa.[41] The reporting was better for childhood onset versus later onset asthma. Pape *et al* found that adults reported quite accurately their parents' smoking during their childhood and their mother' smoking when pregnant with them, when compared with the parents' own report.[42] Timm *et al* found that the accuracy in reporting parental place of upbringing was dependent on own place of upbringing, but this did not bias the associations of place of upbringing with asthma.[43] Skulstad *et al* validated mothers' information about births and pregnancies against data from the Medical Birth Registry of Norway. The analysis found high validity for mother's report of important birth and pregnancy parameters, and that risk-associations were similar when using maternal versus registry-based information.[44]

Life course data on obesity is rarely available for multiple generations. The RHINESSA/RHINE/ECRHS studies have included a tool with pictorial drawings of body silhouettes in childhood, voice break/menarche and adult ages. Dratva *et al* found that current body silhouettes were highly correlated with body mass index (BMI) calculated from measured or self-reported weight and height.[45] Lønnebotn *et al* found that retrospective body silhouettes from adult ages correlated well with BMI calculated from measured height and weight at the corresponding ages in the past, and allowed for differentiation of obesity and non-obesity.[46]

## STRENGTHS AND LIMITATIONS

The main strength of the RHINESSA study is the large number of offspring-parent pairs with rich and similar information from both generations, collected using similar protocols, and with very little missing data on key variables. Both fathers and mothers have been extensively characterised over twenty years of childbearing age, and the availability of such parental exposure information for adolescent and adult offspring is quite unique. The prospectively and retrospectively collected data on family members in this multigeneration study allow validation of information provided about family members,[41–46] thereby extending the number of generations that can be analysed in a robust manner. The multicentre structure is a strength in terms of larger external validity. While the largest number of study participants are from the relatively homogeneous Nordic countries, the Estonian, Spanish and Australian study centres contribute to diversity in the study population improving the external generalisability beyond the Nordic countries. The excellent population and health registries in the Nordic countries represent a major strength of the study, family members can be identified in an unbiased manner and a wealth of data are available for all generations. For some study centres there is information on five generations, covering birth cohorts born over more than one century—the century when the welfare societies were established in many Western societies.

A weakness of the RHINESSA study is that detailed parental data are mostly available for one parent of the offspring, the parent (mother or the father) participating in the ECRHS and RHINE studies. To meet this challenge, a subcohort of the 'other' parents has been studied in Bergen RHINESSA, validation studies have been performed to improve the usefulness of information reported by offspring on both parents, and there are registry data available for both parents in North European study centres. Another weakness is the relatively low response rates. Fortunately, exposure information in terms of parental information is available for responders and non-responders. While selection bias cannot be ruled out, it is reassuring that table 4 suggests similar parental characteristics for responding and non-responding offspring. In study centres with the appropriate parental consent, information on a number of health outcomes in offspring can be obtained from national registries. Finally, the multigeneration multicentre study design is challenging with regard to standardisation of data collection over time and between generations and study centres, and random heterogeneity in the data due to this may attenuate true results. To face this challenge, we used detailed standard operating procedures and coordinated fieldworker training, including extensive interview guides and standardised procedures for translations and back-translations of questionnaires and interviews. The use of self-reported data is encumbered with limitations but key in assessment of respiratory symptoms, occupational titles, etc; fortunately the ECRHS tools are widely used and offer important possibilities to compare with other studies.

So, what lessons have we learnt from the cohort's creation that could be useful for other researchers? One most useful contribution from RHINESSA to other researchers, is the possibility to validate information provided by family members. In general, we find that strong, longstanding collaboration and friendship has been key for creating a complex set of cohorts in a longitudinal multicentre setting. Thus, building on existing cohorts with well-functioning researcher networks appears to be important for future multigeneration epidemiological studies.

**Author affiliations**
[1]Department of Occupational Medicine, Haukeland University Hospital, Bergen, Norway
[2]Centre for International Health, University of Bergen Department of Global Public Health and Primary Care, Bergen, Norway
[3]Department of Clinical Science, University of Bergen, Bergen, Norway
[4]Oral Helath Centre of Expertise Western Norway, Bergen, Norway
[5]Allergy and Health Unit, School of Population and Global Health, University of Melbourne, Melbourne, Victoria, Australia
[6]Medical Faculty, University of Iceland, Reykjavik, Iceland
[7]Department of Sleep, Landspitali University Hospital Reykjavík, Reykjavik, Iceland
[8]Section of Sustainable Health, Department of Public Health and Clinical Medicine, Umeå Universitet, Umeå, Sweden
[9]Occupational and Environmental Medicine, School of Public Health and Community Medicine, Institute of Medicine, Sahlgrenska Academy, University of Gothenburg, Goteborg, Sweden
[10]Lung Clinic, Tartu University Hospital, Tartu, Estonia
[11]Allergy and Lung Health Unit, School of Population and Global Health, University of Melbourne, Melbourne, Victoria, Australia
[12]Department of Medical Sciences: Clinical Physiology, Uppsala University, Uppsala, Sweden
[13]Servicio de Neumología, Complejo Hospitalario Universitario de Albacete, Albacete, Spain
[14]Facultad de Medicina, Universidad de Castilla-La Mancha - Campus de Albacete, Albacete, Spain
[15]Department of Nursing, University of Huelva, Huelva, Spain
[16]Department of Regional Health Research, University of Southern Denmark, Odense, Denmark
[17]Research Unit, Kolding Hospital, University Hospital of Southern Denmark, Kolding, Denmark
[18]Department of Medical Sciences: Respiratory, Allergy, Sleep Research, Uppsala University, Uppsala, Sweden
[19]Department of Obstetrics and Gynecology, Haukeland University Hospital, Bergen, Norway
[20]Department of Public Health - Work, Environment and Health, Danish Ramazzini Centre, Aarhus Universitet, Aarhus, Denmark
[21]National Research Centre for the Working Environment, Kobenhavn, Denmark

**Acknowledgements** A special thanks to all RHINESSA and RHINE/ECRHS study participants, their contributions are greatly appreciated. We also thank the fieldworkers of all the study centres, in particular Nina Særvold, Hilda Andersen and fieldworkers of the Research Unit for Health Surveys, University of Bergen, who had a leading role in the field work of the RHINESSA study.

**Collaborators** RHINESSA International Study Group. Francisco Javier Callejas, Raúl Godoy, JM-M, RJB, Trude Duelien Skorge, Christine Drengenes, AJ, OJ, Maryia Khomich, Jorunn Kirkeleit, Toril Mørkve Knudsen, Ingrid Kuiper, Juan Pablo López-Cervantes, Marianne Lønnebotn, Shokouh Makvandi-Nejad, Antonio Pérez, FGR, Anders Røsland, Rajesh Shigdel, Svein Magne Skulstad, Torgeir Storaas, CS, Øistein Svanes, Kai Triebner, Gro Tjalvin, Hilde Vindenes, Shanshan Xu, MH, Kjell Torén, JLS-R, Jose Maldonado, BB, TG, SD, CJL, Michael Abramson, Lyle Gurrin, Adrian

Lowe, David Martino, Gita Mishra, Jennifer Perret, Bruce Thompson, OJ, Rain Jõgi, AO, LB, Bertil Forsberg, Karl Franklin, AM, CJ, Dan Norbäck, Eva Lindberg, Magnus Ekstrøm, VS, Christine Cramer, Kathrine Pape, Torben Sigsgaard, ST, Anne Mette Lund Würtz, Simone Accordini, Lucia Calciano, Julia Dratva, Joachim Heinrich, John Holloway, William Horsnell, Deborah Jarvis, Susanne Krauss-Etchmann.

**Contributors** The Cohort Profile manuscript was written by CS and VS, AJ had particular responsibility for the tables in addition to revision of the text, all coauthors contributed to discussions during the development of the manuscript and revised the manuscript. CS is principal investigator (PI) of the study; VS is vice PI of the study and PI of Arhus study centre; AJ is lead data manager of the study; FGR is PI for the RHINESSA women studies; CJ is PI of the RHINE study; RJB, SD, BB, LB, MH, OJ, AM, JM-M, AO and JLS-R are current or previous centre PIs of the study; TG, CJL and ST are centre vice PIs. FGR and VS are equal last authors. CS is the guarantor and has full responsibility for the work, had access to the data, and controlled the decision to publish.

**Funding** Coordination of the RHINESSA study has received funding from the Research Council of Norway (Grants No. 274767, 214123, 228174, 230 827 and 273838), ERC StG project BRuSH #804199, the European Union's Horizon 2020 research and innovation programme under grant agreement No. 633 212 (the ALEC Study WP2), the Bergen Medical Research Foundation, and the Western Norwegian Regional Health Authorities (Grants No. 912011, 911 892 and 911631). Study centres have further received local funding from the following: Bergen: the above grants for study establishment and co-ordination, and, in addition, World University Network (REF and Sustainability grants), Norwegian Labour Inspection, the Norwegian Asthma and Allergy Association and Trond Mohn Foundation (Grant ID BFS2017TMT02). Albacete and Huelva: Sociedad Española de Patología Respiratoria (SEPAR) Fondo de Investigación Sanitaria (FIS PS09). Gøteborg, Umeå and Uppsala: the Swedish Heart and Lung Foundation, the Swedish Asthma and Allergy Association. Reykjavik: Iceland University. Melbourne: Australian National Health and Medical Research Council (NHMRC) Project Grant ID1128450. Tartu: the Estonian Research Council (Grant No. PUT562). Århus: the Danish Wood Foundation (Grant No. 444508795), the Danish Working Environment Authority (Grant No. 20150067134), Aarhus University (PhD scholarship).The RHINE study received funding by Norwegian Research Council, Norwegian Asthma and Allergy Association, Danish Lung Association, Swedish Heart and Lung Foundation, Vårdal Foundation for Health CareHealthcare Science and Allergy Research, Swedish Asthma and Allergy Association, Icelandic Research Council, and Estonian Science Foundation. The RHINE IV ongoing study has received funding from the Research Council of Norway project Life-GAP grant No. 300 765.Co-ordination of the ECRHS study has received funding from the European Union's Horizon 2020 research and innovation programme under grant agreement No. 633 212 (the ALEC study), the Medical Research Council (ECRHS III) and the European Commission FP5 and FP7 (ECRHS I and II). The ECRHS IV ongoing study in the ten RHINESSA study centres has received funding from the European Union's Horizon 2020 research and innovation programme projects EPHOR under grant agreement No. 874 703 and European Research Council (ERC) project BRuSH under grant agreement No. 804199, and from the Research Council of Norway grant No. 273 838. Funding agencies for ECRHS I, II and III are reported in online supplement.

**Disclaimer** The funding agencies have had no direct role in the conduct of the study or the data collection and management, nor of data analysis or manuscript preparation.

**Competing interests** None declared.

**Patient and public involvement** Patients and/or the public were involved in the design, or conduct, or reporting, or dissemination plans of this research. Refer to the Cohort Description section for further details.

**Patient consent for publication** Not applicable.

**Ethics approval** This study involves human participants and was approved by Sweden (multicentre: Uppsala, Umeå andGöteborg) screening Regional Ethical Review Board in Uppsala Dnr 2013/ 352 Sweden (multicentre: Uppsala, Umeå and Göteborg) clinical Regional Ethical Review Board in Uppsala Dnr 2016/ 023 Melbourne, Australia Alfred Hospital Human Research Ethics Committee HREC/ 17/ Alfred/ 144 Iceland, Reykjavik questionnaire The National Bioethics Committee VSN-13-190 Iceland, Reykjavik clinical The National Bioethics Committee VSN-16-070 Denmark Ethical Scientific Committee for Mid Region Jylland, Denmark 1-10-72-301-15 Spain, Huelva Comité de ética de la investigación de la provinciade Huelva (Research Ethics Committee of the Province of Huelva) Approval date April 4, 2013 Spain, Albacete Comité ético de investigación clínica del Complejo Hospitalatio Universitario de Albacete. (Ethic Committee of clinical Research of the University Hospital Complex of Albacete)Institutional Review Board IRB 00006998

Estonia,Tartu Research Ethics Committee of the University of Tartu (UT REC) 233/ T-7Norway, Bergen Regional Committee of Medical and Health Research Ethics, Rec West 2012/1077. Participants gave informed consent to participate in the study before taking part.

**Provenance and peer review** Not commissioned; externally peer reviewed.

**Data availability statement** Data are available on reasonable request. Requests for access to data can be made to the RHINESSA steering committee by PI Professor CS (cecilie.svanes@uib.no) or vice PI VS (vs@ph.au.dk). Reuse of the data must be done in collaboration with the RHINESSA study team. Further information including issues on data security and sharing of data can be found at www. rhinessa.net.

**ORCID iD**
Christer Janson http://orcid.org/0000-0001-5093-6980

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
