## [Reviewer comments · BMJ Open]

ARTICLE DETAILS

TITLE (PROVISIONAL)	Cohort profile: The multi-generation Respiratory Health in Northern Europe, Spain and Australia (RHINESSA) cohort
AUTHORS	Svanes, C; Johannessen, Ane; Bertelsen, Randi; Dharmage, Shyamali; Benediktsdottir, Bryndis; Bråbäck, Lennart; Gislason, Thorarinn; Holm, Mathias; Jøgi, Oskar; Lodge, Caroline J.; Malinowski, Andrei; Martinez-Moratalla, Jesus; Oudin, Anna; Sánchez-Ramos, José Luis; Timm, Signe; Janson, Christer; Real, Francisco Gomez; Schlünssen, Vivi

VERSION 1 – REVIEW

REVIEWER	Leventakou, V. Univ Crete
REVIEW RETURNED	12-Dec-2021

GENERAL COMMENTS	There is no doubt that prenatal and birth cohort studies, and those that start before conception, provide an excellent opportunity to investigate how early life exposures may affect later life developmental outcomes and adult disease. The paper is not suitable for publication in its current form. In general, it is very long, and it should be reviewed for brevity to help the reader follow. The authors will need to address some major points and have the option to submit a revised version. The matters to be addressed can be found below. Major Comments: Introduction: - The authors need to make introduction easier and clearer for the reader to follow. My suggestion is to start with the background and not the objective of the proposed study. They should build up their hypothesis, describe what is available (there are other available pre-natal/ pre-conception cohorts), why is this study is important. Introduction can be more condense and focused. - Page 8: the authors mention "...and there are few cohorts that have the needed data". Which studies they refer to? - Page 8: "Study centres in Estonia with Baltic diet and recent transition from middle- to high-income economy, Spain with Mediterranean characteristics, and Australia with particularly high allergy prevalence, contribute to greater generalisability of study results." These characteristics cannot justify the generalizability of the findings. What do the authors mean by 'Mediterranean characteristics' (for Spain)? This is very broad. I would delete this sentence or improve the rationale for the selection of these countries. Cohort description:
--

	 - For all the countries included in the study the national registries were fully operating since the 90's? What about the information for the grandparents reported in Table 4? - Why are parents divided into 2 age groups (Table 4)? Could you please elaborate? - Any sociodemographic characteristics for Table 3? - In Table 5 the missing data (%) refer to the total? Where the same questionnaires used to collect the baseline data across the different centers? - I would suggest combining the 'Participant and public involvement' and the 'Collected data and characteristics of study participant' and make this part briefer. Findings to date:  - This section is very long. A shorter description for these findings (already published elsewhere) would be preferable. The authors could use a table to include the key publications with the appropriate references. They can better group the studies based on exposures, outcomes, validation studies or timepoints. Strengths and limitations:  - Page 27: the authors mention as a strength "...high-quality information from both generations." How is this supported? Please add references. - The large number of participants is undoubtedly a strength, however this is a study that includes heterogenous data sources (different cohorts, languages, concepts etc.) for both the retrospective and the prospective collected data. There is no mention by the authors how they managed to harmonize these data or how they intend to harmonize the new data collected. They acknowledge this as a challenge (limitations, page 28) but do they intend to do something to face this challenge? - For RHINESSA baseline data collection the protocols used were standardized? - Page 28: "...validation studies have been performed to improve the usefulness of information". What type of studies the authors refer to? Add refs. - Page 28: "...homogeneous Nordic countries, the Estonian, Spanish and Australian study centres contribute to the diversity in the study population improving the external generalisability". Generalizability where? to Nordic countries? Spain is a Mediterranean country. Health registries were built up for these countries that long ago that allowed to retrieve info for all generations? - Self-reported data should also be acknowledged as a limitation.
--	---

REVIEWER	Golding, Jean University of Bristol, Centre for Child and Adolescent Health
REVIEW RETURNED	14-Dec-2021

GENERAL COMMENTS	The RHINESSA cohorts are complex, but important resources for all scientists interested in intergenerational and transgenerational associations in humans. There are very few data sets that can address any questions concerning the possible impact of an environmental exposure in one generation affecting their grandchildren or even great-grandchildren. However, although I am delighted in the fact that this manuscript is attempting to define the different groups in the various countries, I still found myself confused. I do not think it would need too much
---

	effort to clarify the structure further. Figures 1 and 2 were very helpful. I suggest that the G1, G2, G3, G4 nomenclature of Figure 2 be used in the text and tables as well. Hopefully this will make the text and tables easier to understand. An additional question concerns to what extent all the participant groups are mutually distinct. Here a few Venn diagrams might be appropriate. The Strengths and Difficulties section is comprehensive and clear. More minor suggestions are as follows: a) In the Abstract, ECRHS is referred to without detailing what the letters stand for. b) Also in the Abstract, 10 centres are referred to in 7 countries – obviously some countries have more than one centre – perhaps you could indicate which by putting the number of centres in brackets after each country – e.g. Norway (3), Sweden (1), c) From Table 3 it appears that the Clinical cohorts have a greater proportion of participants with asthma than the Questionnaire cohorts. I may have missed this, but I did not see that this had been commented upon in the text, either to the reason for it, or the consequences. d) In the heading to Table 5: 2a) and 2b) are referred to. Presumably these should be 'a)' and 'b)'.
--	--

VERSION 1 – AUTHOR RESPONSE

Reviewer: 1

Dr. V. Leventakou, Univ Crete

Comments to the Author:

Comment (C) 1. There is no doubt that prenatal and birth cohort studies, and those that start before conception, provide an excellent opportunity to investigate how early life exposures may affect later life developmental outcomes and adult disease. The paper is not suitable for publication in its current form. In general, it is very long, and it should be reviewed for brevity to help the reader follow. The authors will need to address some major points and have the option to submit a revised version. The matters to be addressed can be found below.

Response (R) 1. The paper has been somewhat shortened, and we believe the changes suggested by the reviewers has made the manuscript considerably easier to follow.

Major Comments:

Introduction:

C2. The authors need to make introduction easier and clearer for the reader to follow. My suggestion is to start with the background and not the objective of the proposed study. They should build up their hypothesis, describe what is available (there are other available pre-natal/ pre-conception cohorts), why is this study is important. Introduction can be more condense and focused.

R2. Thank you, the introduction has been restructured, condensed and better focused – we believe this is now clearer for the reader to follow. Further, other cohorts with main focus on preconception data are mentioned, while cohorts with main focus on prenatal data and early life are considered beyond the scope of this paper for the sake of focus and brevity.

C3. Page 8: the authors mention "...and there are few cohorts that have the needed data". Which studies they refer to?

R3. Examples of cohorts that have the needed data are given in the revised introduction.

C4. Page 8: "Study centres in Estonia with Baltic diet and recent transition from middle- to high-income economy, Spain with Mediterranean characteristics, and Australia with particularly high allergy prevalence, contribute to greater generalisability of study results." These characteristics cannot justify the generalizability of the findings. What do the authors mean by 'Mediterranean characteristics' (for Spain)? This is very broad. I would delete this sentence or improve the rationale for the selection of these countries.

R4. This sentence has been deleted from the revised introduction, as suggested. In the description of the study design, the corresponding topic has been reworded to "...Study centres in Estonia with recent transition from middle- to high-income economy, Spain as a southern European country, and Australia with particularly high allergy prevalence, extend the generalisability of study results beyond Northern Europe where most study centres are situated."

Cohort description:

C5. For all the countries included in the study the national registries were fully operating since the 90's? What about the information for the grandparents reported in Table 4?

R5. There are most registry data available for the very youngest, however, the dates of establishment for the registries from the Nordic countries range from the 18th until the 20th century, and there are useful information for all the generations. In the revised manuscript, this and Table 4 is more clearly described, "In addition, national health registries in the Nordic countries with excellent coverage provide an additional data source for the generations G1-G4 and their family members. Some registries date back to the 18th century, while there are most registry data available the last decades."

C6. Why are parents divided into 2 age groups (Table 4)? Could you please elaborate?

R6. The parents were studied at three study waves, this is now explained in the table heading of the revised manuscript, "For the G2 generation, information from three study waves are presented". Also the consistent use of G1-G4 terminology we believe make this table easier to understand.

C7. Any sociodemographic characteristics for Table 3?

R7. Thank you for the suggestion, information on educational level in three categories, in the mother and father separately, has been added to Table 3.

C8. In Table 5 the missing data (%) refer to the total? Where the same questionnaires used to collect the baseline data across the different centers?

R8. Yes, the missing data refer to the total. This column has been moved to be placed side by side with the column on total, so that this is clear in the revised Table 5. And yes, the same questionnaires were used across different study centres and generations, this is made more clear in the revised manuscript: "The same study protocols (adapted to age) were used in all study centres and all generations, and detailed standard operating procedures (see www.rhinessa.net), interview guides and procedures for translations/back translations contribute to secure harmonisation of data across study centres and generations."

C9. I would suggest combining the 'Participant and public involvement' and the 'Collected data and characteristics of study participant' and make this part briefer.

R9. Since these subheadings are specifically asked for by the journal, we have kept these, but the "participant and public involvement" part has been shortened to the half while still keeping the information asked for by journal guidelines.

Findings to date:

C10. This section is very long. A shorter description for these findings (already published elsewhere) would be preferable. The authors could use a table to include the key publications with the appropriate references. They can better group the studies based on exposures, outcomes, validation studies or timepoints.

R10. The text has been shortened to about two thirds, and a Table 6 with key publications has been added.

Strengths and limitations:

C11. Page 27: the authors mention as a strength "...high-quality information from both generations." How is this supported? Please add references.

R11. While certain parts of the ECRHS questionnaire have been validated, this is difficult to support, and the wording has been altered to "The main strength of the RHINESSA study is the large number of offspring-parent pairs with rich and similar information from both generations, collected using similar protocols, and with very little missing data on key variables".

C12. The large number of participants is undoubtedly a strength, however this is a study that includes heterogeneous data sources (different cohorts, languages, concepts etc.) for both the retrospective and the prospective collected data. There is no mention by the authors how they managed to harmonize these data or how they intend to harmonize the new data collected. They acknowledge this as a challenge (limitations, page 28) but do they intend to do something to face this challenge?

R12. Thank you, we have added "To face this challenge, we used detailed standard operating procedures and co-ordinated field-worker training, including extensive interview guides and standardised procedures for translations and back-translations of questionnaires and interviews."

C13. For RHINESSA baseline data collection the protocols used were standardized?

R13. Yes, and thank you for noting this information was not clear. The following has been added to make this clear in the revised manuscript "The same study protocols (adapted to age) were used in all study centres and all generations, and detailed standard operating procedures (see www.rhinessa.net), interview guides and procedures for translations/back translations contribute to secure harmonisation of data across study centres and generations."

C14. Page 28: "...validation studies have been performed to improve the usefulness of information". What type of studies the authors refer to? Add refs.

R14. References has been added.

C15. Page 28: "...homogeneous Nordic countries, the Estonian, Spanish and Australian study centres contribute to the diversity in the study population improving the external generalisability". Generalizability where? to Nordic countries? Spain is a Mediterranean country. Health registries were built up for these countries that long ago that allowed to retrieve info for all generations?

R15. This sentence has been reworded, "...improving the external generalisability beyond the Nordic countries". Concerning health registries, yes, information is available for all generations and this is now better described, see response to C5.

C16. Self-reported data should also be acknowledged as a limitation.

R16. This has been added.

Reviewer: 2

Dr. Jean Golding, University of Bristol

Comments to the Author:

The RHINNESSA cohorts are complex, but important resources for all scientists interested in intergenerational and transgenerational associations in humans. There are very few data sets that can address any questions concerning the possible impact of an environmental exposure in one generation affecting their grandchildren or even great-grandchildren.

C1. However, although I am delighted in the fact that this manuscript is attempting to define the different groups in the various countries, I still found myself confused. I do not think it would need too much effort to clarify the structure further. Figures 1 and 2 were very helpful. I suggest that the G1, G2, G3, G4 nomenclature of Figure 2 be used in the text and tables as well. Hopefully this will make the text and tables easier to understand.

R1. Thank you very much. In the revised manuscript we are using this suggested nomenclature consistently throughout the manuscript, and we believe this has made text and tables much easier to understand.

C2. An additional question concerns to what extent all the participant groups are mutually distinct. Here a few Venn diagrams might be appropriate.

R2. An illustration showing the relationship between the questionnaire and clinical study phases are now added as an online supplement to the manuscript. We would be happy to have this as part of the main manuscript, if that is desirable.

The Strengths and Difficulties section is comprehensive and clear.

More minor suggestions are as follows:

Ca) In the Abstract, ECRHS is referred to without detailing what the letters stand for.

Ra) This has been corrected.

Cb) Also in the Abstract, 10 centres are referred to in 7 countries – obviously some countries have more than one centre – perhaps you could indicate which by putting the number of centres in brackets after each country – e.g. Norway (3), Sweden (1),

Rb) This has been added, as suggested.

Cc) From Table 3 it appears that the Clinical cohorts have a greater proportion of participants with asthma than the Questionnaire cohorts. I may have missed this, but I did not see that this had been commented upon in the text, either to the reason for it, or the consequences.

Rc) Thank you for noting this, this is in the revised manuscript explained “Clinical examinations were conducted in subsamples from ~45 study centres, primary random subsamples, but for some centres an additional subsample with persons with asthma symptoms”, and commented on “As expected due to the original sampling strategy in ECRHS (enriched with persons with symptoms) the prevalence of asthma is somewhat higher in the clinical sample compared to the questionnaire sample.”

Cd) In the heading to Table 5: 2a) and 2b) are referred to. Presumably these should be ‘a)’ and ‘b)’.

Rd) This has been corrected.

VERSION 2 – REVIEW

REVIEWER	Leventakou, V. Univ Crete
REVIEW RETURNED	06-Mar-2022

GENERAL COMMENTS	The authors adequately addressed my previous comments. Although I still find this manuscript quite long i have no further comments to add and suggest to be accepted for publication.
---

REVIEWER	Golding, Jean University of Bristol, Centre for Child and Adolescent Health
REVIEW RETURNED	06-Mar-2022
GENERAL COMMENTS	Thank you for clarifying various aspects of this complex study design. This is really important since the contribution